# Telemedicine, E-Health, and Multi-Agent Systems for Chronic Pain Management

Manar Ahmed Kamal [1], Zainab Ismail [2], Islam Mohammad Shehata [3], Soumia Djirar [4], Norris C. Talbot [5], Shahab Ahmadzadeh [6], Sahar Shekoohi [6,*], Elyse M. Cornett [6], Charles J. Fox [6] and Alan D. Kaye [6]

1   Faculty of Medicine, Benha University, Benha 13511, Egypt
2   Faculty of Medicine, Menoufia University, Shiben El Kom 51123, Egypt
3   Department of Anesthesiology, Faculty of Medicine, Ain Shams University, Cairo 11517, Egypt
4   Department of Ophthalmology, University of Montreal, Montreal, QC H3T 1J4, Canada
5   School of Medicine, Louisiana State University Health Sciences Center at Shreveport, Shreveport, LA 71103, USA
6   Department of Anesthesiology, Louisiana State University Health Sciences Center at Shreveport, Shreveport, LA 71103, USA
*   Correspondence: sahar.shekoohi@lsuhs.edu

**Abstract:** Telemedicine, telehealth, and E-health all offer significant benefits for pain management and healthcare services by fostering the physician–patient relationship in otherwise challenging circumstances. A critical component of these artificial-intelligence-based health systems is the "agent-based system", which is rapidly evolving as a means of resolving complicated or straightforward problems. Multi-Agent Systems (MAS) are well-established modeling and problem-solving modalities that model and solve real-world problems. MAS's core concept is to foster communication and cooperation among agents, which are broadly considered intelligent autonomous factors, to address diverse challenges. MAS are used in various telecommunications applications, including the internet, robotics, healthcare, and medicine. Furthermore, MAS and information technology are utilized to enhance patient-centered palliative care. While telemedicine, E-health, and MAS all play critical roles in managing chronic pain, the published research on their use in treating chronic pain is currently limited. This paper discusses why telemedicine, E-health, and MAS are the most critical novel technologies for providing healthcare and managing chronic pain. This review also provides context for identifying the advantages and disadvantages of each application's features, which may serve as a useful tool for researchers.

**Keywords:** multi-agent systems; telemedicine; e-health; chronic pain; pain management

## 1. Introduction

Many different technologies have been developed worldwide to create new ways of managing various chronic pain disorders [1]. Electronic-based technologies such as telemedicine, which in many cases is used interchangeably with telehealth, and E-health are essential innovations that facilitate the communication between the healthcare providers and receivers [1,2]. Telemedicine uses communications and information technology to provide healthcare services such as monitoring patient treatment response and counseling to people living away from the hand of healthcare providers [3]. Telemedicine health services can be delivered by call centers, cell phones, videoconferencing, and web-based platforms [1]. Telemedicine also helps in information recording, storing, and giving convenient access of a patient's electronic records to a healthcare provider [1]. In a pilot study, Peng et al. showed that telemedicine-based follow-up of chronic pain patients is beneficial and a money-saver for patients who live in remote areas. Both patients and physicians in the study preferred telemedicine consultation [4].

E-health is a rapidly developing research field that includes applying digital modes such as computers and smartphones to support or present health interventions [5]. E-health programs have several advantages: the accuracy of the processes itself, convenience for patients' time and privacy, and patients avoiding face-to-face interactions [5]. The range of E-health to approach multiple demographics exists in its ability to conduce programs in many different media forms. The various forms of E-health interventions extend from simple text-based programs to complicated, multimedia, and interactive programs such as virtual reality systems [6]. For example, E-health virtual reality interventional systems depend on distraction to reduce pain in children, including different games, interactive musicals or story books, and interactive toys [7]. Moreover, the virtual reality system involves biofeedback programs such as galvanic skin response and heart variability sensors that detect and manage stress-related physiological changes and provide training programs for relaxation [8]. E-health provides excellent opportunities for pain management and healthcare services due to a rapidly growing field of computer programming and technology [1]. An important part of artificial intelligence is the agent-based system that becomes a developing area to deal with simple or complex problems [9].

Multi-Agent Systems (MAS) are well-established modalities to model and solve real-world problems [9]. The key idea of MAS is to create communication and cooperation between agents, which are generally considered computerized autonomous units, to solve various problems [9]. MAS involve many telecommunication applications such as the internet, robotics, and medical applications [9]. MAS and information technology are used to improve the palliative care provided to the patients [10].

The current existence of telehealth, E-health, and MAS has a significant role in creating more efficient and patient-oriented care for chronic pain management. The current state of the world has enveloped itself in technology and innovation, yet healthcare, especially in chronic pain management, has barely appeared to recognize the benefits and improvements of telehealth-, E-health-, and MAS-based treatment plans for patients. While telemedicine, E-health, and MAS have an essential role in chronic pain management, the published studies discussing their role in chronic pain treatment are still insufficient. However, the limited existing literature is pointing to all the signs of their future roles in chronic pain management. This review discusses why telemedicine, E-health, and multi-agent systems are the most critical technologies for providing healthcare and managing chronic pain. It also provides context for identifying the advantages and disadvantages of each application's features, which may serve as a resourceful tool for researchers.

## 2. Background about Multi-Agent Systems

There are several methods to define the "agent" component of MAS [11–13]. Generally, the agent can be defined as an essential autonomous software that functions constantly in dynamic and unpredictable settings, and reacts to events while displaying intelligent behavior to achieve its goals [14]. MAS include many interacting computer units known as agents, which have the following characteristics: autonomy, localized perspective, decentralized control, and data [15]. MAS may alternatively be defined as abstractions capable of encapsulating the core of numerous software systems at various degrees of detail rather than a single technology that enables the implementation of distributed intelligent systems [14]. Agents and MAS are now considered the most significant system levels accessible [16,17]. Both types provide a unique degree of abstraction in analyzing, designing, and implementing large software systems [18]. Importantly, MAS are designed for multiple potential settings in which the agents consist of autonomous entities that respond to the input of robots and humans. Moreover, these systems are designed to take input from multiple areas, whether it be human or autonomous, and then reply with a computational approach based on cross-referencing all the available data.

The components of MAS are a pair (A and Env) where "A" is a set of agents (Ag1, Ag2, . . . etc.), and "Env" is a collection of potential environmental states [19]. Those components may be running on various computers in different locations [10]. Each agent

may retain a portion of the knowledge necessary to solve the problem; hence, MAS provide a natural means of addressing dispersed challenges [10]. Modalities of MAS for healthcare usage are seen in the agent's corresponding information about a patient from multiple informational stockpiles. In Figure 1, an MSA system example is shown to illustrate the computational architecture of such a system.

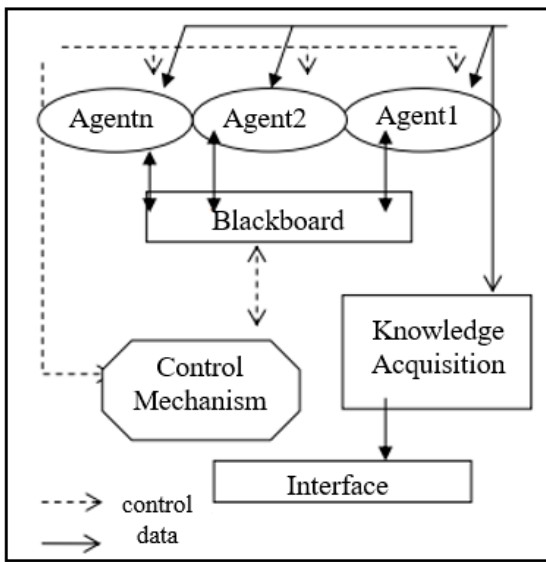

**Figure 1.** MAS example architecture [20].

Representing the multiple automated agents that actively incorporate live data, this diagram shows the agent slots' ability to interact with the larger system and each other. The blackboard serves as the environment variable and allows the agents to quickly adapt to incoming information [20]. The MAS potential abilities are discussed later; however, it is important to note that MAS change computerized remote health monitoring into an intricate system that ciphers through all possible health data to help physicians make decisions faster. They reduce the work load of healthcare workers and integrate data from multiple specialties about a single patient to make more informed decisions [21]. MAS have been characterized with the following properties: sociability, pro-activeness, autonomy, and reactivity [22]. Sociability is one of the essential properties by which the agents can communicate between themselves, use agent communication language, and exchange information [10]. Through these means, any agent can provide and ask for services [22]. Proactiveness is another essential property of agents defined as performing tasks that may benefit the user, even if the user has not explicitly demanded those tasks be executed. The agent may exhibit goal-directed behavior by using a pro-activeness property to find relevant information and show it before the user requests it [10]. For example, the agent knows that the user is traveling abroad, so the agent will look for information about the medical centers in town as containing a cardiology department if it knows that the user has had any previous heart problems and might need this information urgently [10]. The essential property of any agent is its autonomy. The agent can possess individual goals and self-resources without any direct human or other intervention. The agent has some degree of control over its actions and internal state and regulates its functions without outside assistance or supervision [22]. Autonomy involves the agent's ability to make decisions based on its internal state and its information from the environment [10].

Reactivity is the last property in which the agent can perceive and act in a close circuit. Moreover, it can respond in a timely fashion to the changes around it [22]. Therefore, the fundamental properties of MAS and their distinguishable features (management of distributed information, communication, and coordination between separate autonomous entities) present a competent strategy to consider on solving the problems in the healthcare domain.

### 3. Multi-Agent Systems and Telehealth

Telemedicine and telehealth are the structured methods of delivering care to hard-to-reach patients. Telemedicine refers to providing healthcare services over distance using information and communication technologies, and it is a promisingly developing technology across the world [23]. However, E-health has described a broader range of digital technologies and interventions used by various stakeholders across diverse settings [2]. Telehealth services include remote consultation with specialists; presenting a diagnosis, treatment plan; giving medical advice to the patients with remote access to preventive services; distance learning for continuing medical education; and telemonitoring to redefine treatment plans [2,24]. All these medical services have been provided in a timely and convenient method regardless of the location of the patients. The multi-agent autonomic system includes autonomic managers and intelligent control loops, which are core elements of the autonomic systems [24]. The intelligent control loops deal with different events in the system, make all components easy to communicate, collaborate, and enable high-level management tools to regulate themselves and mask inherent system complexities from users. Every control loop includes monitoring, analyzing, planning, and executing the ordered tasks [25]. Additionally, the multi-agent autonomic system architecture includes reactive software, coordinating software, deliberative software, and telehealth information and knowledge management subsystems [24].

The current limitations of chronic pain management are directly envisioned through the patients' struggles. Chronic pain patients can be too severely debilitated or older patients that would extremely benefit from the novel innovation of telehealth [26]. For example, current modalities require follow-up appointments and monitoring, which entails patients possibly enduring uncomfortable trips to see a doctor; however, telehealth offers the ability to avoid strenuous and taxing trips that can further a patient's ailments [26]. Paired with MAS, telehealth can potentially revolutionize the job and workload of a physician to manage patients outside of person-to-person appointments. On top of the more efficient management of patients, telehealth for chronic pain has started to show signs of decreasing costs [27]. One study found lower costs for patients undergoing telerehabilitation for nonspecific chronic back pain [27]. Altogether, telehealth is continuing to prove its usability in chronic pain management, and the addition of MAS would further its potential in treating patients thoroughly and efficiently.

### 4. Telehealth Modalities

Several telehealth technology modalities allow the connection between the healthcare personnel (HCP) and the patients to deliver healthcare services [28]. These modalities include synchronous, asynchronous, and remote patient-monitoring.

Synchronous refers to real-time telephone or live audio–video interaction where typically the patient uses smartphones, tablets, and computers. Peripheral medical equipment such as digital stethoscopes, otoscopes, or ultrasounds can be used physically with the patient by other HCP as nurses or medical assistants. In contrast, the consulting medical provider conducts a remote evaluation [28].

Asynchronous is another modality that refers to store and forward technology where messages, images, and data are collected and interpreted. This secure-messaging communication between the healthcare provider and patient can be facilitated easily depending on patient portals [28]. Remote patient monitoring is the direct transmission of a patient's clinical measurements to the healthcare providers from a distance. These aspects, for example, can relay heart failure monitors, asthma monitors, and tremor monitors [29–31].

The different aspects of telehealth allow a comprehensive analysis of areas of benefit. For example, some studies have shown that nutrition management in older adults living at home is beneficial to patient care, and others can contribute to the diagnosis of something such as diabetic foot ulcers. The advancement of telehealth sees that patients are receiving treatment that in many cases they would not receive otherwise [32]. A quickly growing utilization of technology and remote management of patients allows for a robust organization

of diagnosis and management of patients. Moreover, areas involving chronic pain have massive potential to ease the lives of both patients and healthcare workers [33]. A recent study focused on chronic pain management in veterans to show that telehealth modalities contributed to lower costs, reduced missed appointments, and achieved a high satisfaction rate amongst patients and healthcare workers [34]. The implementation of telehealth for chronic pain appears as the newest tool to reduce costs and treat patients more efficiently. It prevents patients with severe pain from making unnecessary trips and ensures having the appropriate follow-up appointments with their healthcare providers [33].

## 5. Benefits and limitations of Telehealth

Recently, with the advent of the coronavirus disease 2019 (COVID-19), telehealth services have become an established form of healthcare. Telehealth has been helping the public health mitigation measures by limiting possible infectious exposures, making this technology a safer alternative for both HCP and patients. The telehealth strategy can alleviate the burden on healthcare systems by reducing the surge of patient demand on exhausted medical facilities, especially during the recurrent COVID waves. Another economic advantage is reducing the working hours and personal protective equipment (PPE) by healthcare personnel. Allowing remote access to healthcare services should improve participation among medically or socially deprived individuals who do not have easy access to HCP. When an in-person visit is not practical or feasible, remote access can assist in sustaining the patient–provider relationship [28]. Furthermore, telehealth presents usable service for the convenience of the patient in terms of no transportation time or restriction by distance to the physician, exhibiting obvious benefits in terms of preventing patients with chronic pain from making unnecessary trips to an appointment.

Telehealth has become a more common modality that has the potential to revolutionize remote care. The current system permits the working together of multiple different healthcare areas to treat a patient without that patient having to be seen in person. Moreover, healthcare workers, patients, and medical devices can consistently work together to provide up-to-date real care. Comparatively to MAS, telehealth has similar goals in mind to connect multiple areas of patient care. However, MAS are the solution to monitor a patient's health with less work and faster results from healthcare professionals. MAS would provide the autonomous agents and systems that can cross-reference patient data and medical devices while simultaneously conversing potential treatment plans and options for healthcare workers [35]. The addition of MAS into an already growing area of telemedicine generates a massive potential to revolutionize speed and efficiency of patient care away from the hospital [35]. For example, one study found that the utilization of telehealth for treatment of postpartum depression actively reduced feelings of depression and anxiety in women [36]. These findings can lead to the advancement of MAS with telehealth to increase the effectiveness of treatment for chronic pain management by monitoring and controlling decisions based on an active flow of cross-referenced information of a patient's entire medical history.

However, telehealth services have some limitations, such as situations in which in-person visits are preferable due to the complexity of the underlying health problems, the inability to do an adequate physical exam, and patient privacy concerns. Moreover, the inadequate access to a technical smartphone, tablet or computer required for a telehealth appointment, connectivity difficulties, and comfort level with technology for both HCP and patients are more reasons for the delayed growth and openness of telehealth. With an aging population, the electronic literacy capabilities of older generations could be a detriment to efficient and usable telehealth. In the case of technological competency, healthcare providers would also need to learn and adapt to the developments of telehealth, which could be time-consuming and inefficient in some cases. Finally, some patients may not culturally accept virtual visits in place of in-person appointments [28]. While these facts can hinder telehealth usage, more problems arise with the lack of human social interaction

that could potentially contribute to feelings of anxiety or loneliness. An increase in remote healthcare has indications for low levels of person-to-person interaction.

Moreover, telehealth is a constantly growing tool that sees a larger need for healthcare literacy. Healthcare workers would have to be trained and up to date on different modalities and forms of telehealth, which is constantly evolving [37]. While many options exist to train healthcare professionals, unwillingness and difficulty can continue to grow in workers who are not appropriately trained with these tools. A study on telemedicine in cancer outlines all the potential needs and training involved with telehealth to protect patient privacy as well as provide suitable care. In these types of studies, the barrier to learning these new tools serves as a roadblock to efficiently integrate them into patient care [38].

Importantly, telehealth, along with any other form of remote or technological system, comes with its ethical issues and concerns for physicians. Initially, physicians must be aware of the risks to privacy and confidentiality for a patient as well as deciding the best form of care through remote usage. Furthermore, physicians must be aware of any shortcomings in the fidelity and care of the patient, emphasizing competent care over a remote connection. It is also the role of healthcare workers to be transparent and have consent for any telemedicine appointment. The role of the physician in telemedicine as well should be to continue the continuity of care, and healthcare workers should view it as a means to improve and create more efficient care. In a patient-oriented setting, physicians must be aware of the potential shortcoming that telemedicine can cause in consistent and confidential patient healthcare (Table 1) [39].

**Table 1.** Benefits and limitations of telehealth.

| Benefits | Limitations |
|---|---|
| Reduction of the surge of patient demand on the exhausted medical facilities | Complexity of the underlying health problems |
| Reduction of the working hours and PPE | The inability to do an adequate physical exam |
| Assistance in sustaining the patient–provider relationship | Patient privacy concerns |
| Usable service for the conveniency of the patient | Ethical issues and concerns for physicians |
| Permits the working together of multiple different healthcare areas to treat a patient without that patient having to be seen in person | Inadequate access to a technical smartphone, tablet or computer and comfort level with technology |

## 6. Multi-Agent Systems and E-Health

E-health has become a standard modality for the advancement of daily practices of medical clinics. MAS and E-health together allow for better availability of medical services to patients [40]. Moreover, E-health cannot be promoted without a specified MAS in the facility or place of practice. Multi-system healthcare applications can take outstanding advantage of the intrinsic characteristics of MAS because of notable factors. Those systems are critical in applications that remotely monitor patients with chronic ailments or in-home care [41]. Through the past decade, many studies discussing E-health systems in healthcare settings have been generated and synthesized into a set of useable recommendations for clinical practice [42].

The benefits of E-health include cost-effective services with good quality, which encourages governments and healthcare systems to recruit resources for E-health [43]. For example, the internet allows E-health users to communicate with healthcare professionals by email, access medical records, research health information, and engage in a person-to-person exchange of text, audio, video, and other data [42]. The main components of a MAS are interface, manager, information, and intelligent agent [40]. MAS have developed the theory and the practice of high-speed, mission-critical, content-rich, decentralized information systems; this innovation has been achieved by integrating the mutual interdependencies, dynamic environments, and sophisticated controls to play a unique role.

Importantly, E-health has continued to grow in terms of products and tools utilized by patients. For example, a recent study displayed important improvements in chronic pain management for chronic low back pain through the utilization of E-health self-management programs [44]. With a steady increase in E-health usage, it is becoming increasingly more important to implement systems that aid physicians in making accurate medical decisions. The amount of money spent on medical monitoring devices just eclipsed the USD 30 billion mark as seen in Figure 2.

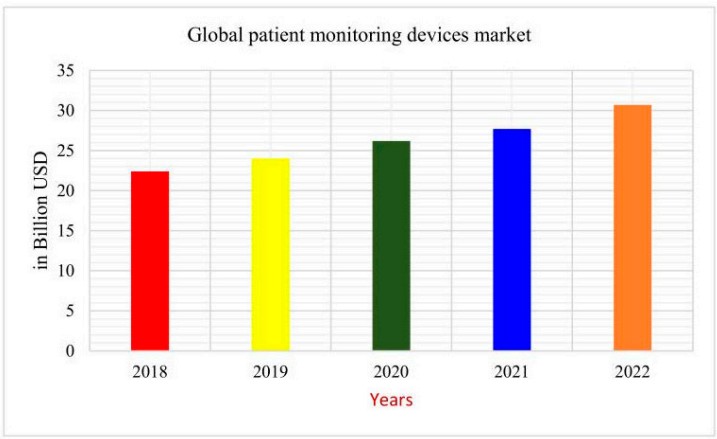

**Figure 2.** Global expenditures for medical monitoring devices [21].

However, each of the devices employed in the computerized side of medicine need a system to quickly respond to patient needs while incorporating multiple fields of information to treat a patient. These monitors are one example of systems that need constant dynamic monitoring to relay wide ranges of information to physicians, especially in the case of emergencies. Many modern mobile devices are designed with personal computing, internet capabilities, and compatibility with the downloadable applications (apps) that instantly access health information [23]. E-health applications are often known as smart-hospital applications; such applications try to improve the daily activities of doctors and nurses [40]. In such specialties and departments, emergency physicians can access the records of their patients in advance while they are still in the ambulance approaching the location of the patient [45].

The utilization of MAS with preexisting E-health modalities is important to fully assess their potential. While MAS have their advantages, their ability to dynamically make suggestions and calculations can be paired with E-health to provide better results for E-health usage. For example, one study explored the potential of remote appointments and analysis of the efficacy of an E-health type visit. The study found weaknesses in the ability to treat patients after a remote visit due to geographical differences [46]. However, this study permits a perfect example of the operation of MAS with modern E-health symptoms to overcome these weaknesses. MAS can provide an autonomous calibration of data to make informed decisions. This system paired with E-health can determine the efficacy of remote visits and distinguish the need or potential outcomes for each individual patient. This system would allow E-health tools to be accessed more appropriately while creating an environment for the best possible care. Moreover, in a hybrid scenario MAS can determine a series of suggestions for physicians to consider which options are more suitable at different points in the patient's care. The immediate cross-referencing of information has the potential to refer a patient for in-person appointments, remote appointments, and medicine scheduling, and can also propose responses in the case of an emergency for a patient.

Furthermore, one study even tested the utilization of vital-signs testing paired with a hospital database agent to relay information to specialists about patients remotely. The study found it beneficial in older patients, remote patients, and chronic patients [21]. Uti-

lizing a hospital-based agent that read the vital signs, the system was able to compile information and found usable data that covered a patient's medical history along with readings important for multiple specialties. These MAS are capable of accessing the hospital databases as well to compile information on a patient in order to aid physicians in making informed decisions [21]. The systems are the cutting-edge technology of the future to decrease errors and utilize all possible information. This growing technology permits flexibility and the interlink of multi-systems and E-health use. However, there is a continuing need for more extensive telemedicine studies, for example, controlled interventions that focus on patients' perspectives, economic analyses, and telemedicine innovations with complex processes and ongoing collaborative achievements. Moreover, different factors should be considered, such as privacy, reusability, modularity, personalization, system maintenance, and studied carefully in the following years to bring agent technology to practical settings. While society continues to develop new and improved technologies, the utilization and understanding of E-health to further improve patient care is a constantly growing necessity. E-health provides the innovation and excitement of the future to benefit both patients and healthcare workers.

## 7. Future Directions

Management of chronic pain disorders includes screening, diagnosis, treatment, and follow-up of the patient's response. Telemedicine and MAS proved promising benefits in most of these steps, which may ensure proper management. In 2016, the Centers for Disease Control and Prevention (CDC) reported that more than 2 out of 10 United States citizens experienced chronic pain at some point in their lives, and 8 out of 10 citizens had high-impact chronic pain [47]. The number of senior adults is expected to grow by 56% by 2032 compared to 2015, which may present an economic burden [48]. One major goal of MAS is to enhance the older adults' lifestyle with a financially proper budget [49]. MAS can provide a virtual world that connects healthcare providers, social assistants, patients, and relatives to ensure the best treatment plan possible [14]. Nanotechnology offers even smaller body sensors that are either wearable or implantable, together with a wireless sensor network from a multi-agent distributed platform (MADIP). The MADIP can perform continuous biochemical monitoring, detect and predict various abnormalities, provide remote assistance, and send feedback to the responsible caregivers [1]. Many commercial products using MAS are now available such as fall detectors, gate analytics, and electrocardiograms (ECG) [50]. These updated technologies are notable for their affordability, enhanced battery life, and new energy resources such as airflow, heat engines, and solar energy [51]. Furthermore, advancing technologies in remote monitoring of heart failure, tremors, digital inhalers, and asthma are some of the many promising E-health innovations that can be paired with MAS for quick and consistent patient care [29–31].

## 8. Virtual Reality and Chronic Pain Management

Virtual reality uses 2-dimensional and 3-dimensional technology to improve the interaction between patients and physicians. Virtual reality is subdivided into augmented reality, haptic sensation, and 3-dimension hologram [33]. Many studies have proved that augmented reality could increase the pain-threshold level and range of motion, shorten the rehabilitation, and decrease pain intensity and distress [2]. Although high-definition 3D (HD3D) conferences virtually engage the patient and physician, physical existence and examination remain absent, which may be a critical component in making a proper diagnosis. Haptic technology can add the touch element, making it possible to palpate tenderness, and evaluate muscle tone and strength and cutaneous sensation [52,53]. Moreover, augmented reality and haptic technology keep recording the results in acute and chronic pain management and patients' rehabilitation, which may motivate the patient to complete the therapeutic course. Virtual reality recorded lower rates of anxiety and a lower sensitivity to pain for children aged 6 months to 18 years. With the use of virtual reality,

children were more likely to have an easier time with procedures and visits, leading to less healthcare avoidance as an adult [54].

### 9. Telemedicine, E-Health, and Chronic Pain Management

Telemedicine has shown cost-effective benefits for chronic pain management in many areas worldwide, but some areas are struggling to adapt with the necessary resources [4]. Although the American Telehealth Association was established in 1993, most developing countries have not established a telemedicine infrastructure [49]. The infrastructure involves a good internet network, phone settings, webcams, up-to-date IT units, and affordable access. Telemedicine still faces ethical and social constraints in some conservative societies. The COVID-19 pandemic encouraged many countries to embrace telemedicine as an alternative to conventional practice. McGill University health center set an excellent example for merging telemedicine with pain management. They had four telemedicine units for pain management before the COVID-19 pandemic. They have been developing a transition to a low-budget system independent from IT units that ensures patients' confidentiality [33]. On the other hand, in developing countries such as Egypt, HCP have invented simple online communication methods to provide good health services to their patients [55].

Certain studies have found the usage of telehealth modalities to benefit pain management. One such study by Hernando-Garijo et al. found that women placed into a telerehabilitation program saw improved benefits in terms of pain sensitivity, mechanical pain sensitivity, and psychological distress [55,56]. Another study implemented mobile health apps to improve patient health and wellbeing with chronic low back pain [56]. The study found patients were receptive to these processes, and it decreased the rate at which patients sought further care for their pain. Furthermore, the addition of telehealth into patient care saw serious health problems and pain management more properly managed. A recent study found that telehealth had a statistically significant effect on pain severity and pain interference in patients with cancer pain [57]. This study found a higher efficacy of care through telehealth modalities rather than just the classical managements [58]. The addition of telehealth into chronic pain management has seen a steady increase in patient satisfaction and results for chronic pain management [59]. Galiano-Castillo et al. found similar results pertaining to telehealth usage for breast cancer survivors, utilizing internet-based exercise programs that were found to improve health status, physical, role, cognitive functioning, and arm symptoms in patients. Telehealth in this manner permitted the studying of different conditions and improved approaches for the management of pain [60]. Another study by Bailey et al. saw an average improvement in pain as well as decreased levels of depression and anxiety due to the utilization of a digital care program for patients [32]. Altogether, these studies provided positive outlooks into the future of telehealth modalities for chronic pain while also suggesting potential long-term benefits and management of chronic pain.

### 10. Other Efforts Regarding Chronic Pain Management

A new educational E-health tool using patients' clinical data and psychological factors to properly select patients eligible for spinal-cord stimulation surgeries is now available on https://www.scstool.org (accessed on 1 February 2023) [61]. A pilot randomized controlled trial based on a chatbot using cognitive therapy for pain management called "self-management of chronic pain" (SELMA) is showing a positive outcome [62]. Additionally, robots may facilitate patients' movement and rehabilitation due to their ability to sense and analyze the data from patients and their surroundings [63]. However, this necessitates a massive collaboration from biomedical engineering companies to develop medical robots. It is imperative to understand that telemedicine and artificial intelligence can revolutionize chronic pain diagnosis and treatment. However, as in any emerging field, both economic evaluation and further studies discussing the long-term outcomes should be conducted.

## 11. Conclusions

Telemedicine, E-health, and MAS have achieved promising results with acute and chronic pain management. Telemedicine is successfully used in counseling, monitoring, and follow-up, especially for senior and disabled patients living away from healthcare centers. Recently, as evident with the COVID-19 pandemic, telemedicine helps foster communication between patients and physicians while maintaining social distancing. Telehealth reduces the need for travel and can save money and time as well as preventing sick patients from interacting with other ill patients. Telemedicine still requires a challenging development journey, especially in developing counties where the competent infrastructure is still missing. In addition, both medical staff and patients need intensive training in telecommunication to save the time consumed with technical problems. It also removes some of the personal aspects of meeting with doctors in person.

On the other hand, E-health facilitates telemedicine use in patients' management by granting physicians access to a patient's data and biochemical and radiological investigations all in one click, which saves both time and effort. Moreover, new E-health tools can help physicians incorporate the patients and their families to decide the best approach for management. The utilization of MAS in a healthcare setting has amazing potential to reduce errors and provide more successful clinical outcomes. Furthermore, the use of these systems permits faster and more intricate responses for doctors to compare a multitude of data that were cross-analyzed from a patient's medical history. Many advanced technologies such as virtual reality are now evolving to ensure the proper diagnosis, intervention, and best rehabilitation plan. MAS have some properties that can also open new doors in healthcare, such as autonomy, sociability, reactivity, and pro-activeness, which ensure a more comprehensive way of management depending on the interaction between different agents. Telemedicine and E-health combined with artificial intelligence can revolutionize chronic pain management, necessitating further research investigating the long-term impact. With more studies tailored to these highly promising methods, healthcare has the potential to be revolutionized by technology for both patient care and safety as well as easing the burden on healthcare workers.

**Author Contributions:** Study concept and design, M.A.K., Z.I., I.M.S., S.D., N.C.T., S.A., S.S., E.M.C., C.J.F. and A.D.K.; analysis and interpretation of data, M.A.K., Z.I., I.M.S., S.D., N.C.T., S.A., S.S., E.M.C., C.J.F. and A.D.K.; drafting of the manuscript, M.A.K., Z.I., I.M.S., S.D., N.C.T., S.A., S.S., E.M.C., C.J.F. and A.D.K.; critical revision of the manuscript for important intellectual content, M.A.K., Z.I., I.M.S., S.D., N.C.T., S.A., S.S., E.M.C., C.J.F. and A.D.K.; statistical analysis, M.A.K., Z.I., I.M.S., S.D., N.C.T., S.A., S.S., E.M.C., C.J.F. and A.D.K. All authors listed have made a direct and intellectual contribution to the work and approved for publication. All authors have read and agreed to the published version of the manuscript.

**Funding:** This research received no external funding.

**Institutional Review Board Statement:** Not applicable.

**Informed Consent Statement:** Not applicable.

**Data Availability Statement:** Data sharing is not applicable to this article as no datasets were generated or analyzed during the current study.

**Conflicts of Interest:** The authors declare no conflict of interest.

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
