# Peer review of "Telemedicine, E-Health, and Multi-Agent Systems for Chronic Pain Management"

_clinpract, doi:10.3390/clinpract13020042_

Round 1

Reviewer 1 Report

This paper highlights the benefits of telemedicine, telehealth, and e-health in pain management and healthcare services, especially in challenging circumstances. It also addresses the role of Multi-Agent Systems (MAS) in these fields, which is rapidly evolving as a solution to various real-world problems. MAS and information technology are used to enhance patient-centered palliative care and are considered critical technologies in managing chronic pain. However, the published research on their use in treating chronic pain is currently limited.

Although the paper provides a useful overview of the advantages and disadvantages of each technology, I have a few suggestions for how the text could be improved. In brief, some possibilities include making modifications to the language, adding additional information (e.g., references), clarifying certain points, or making the text more engaging and visually appealing. By implementing these suggestions, the text will become more effective in conveying its message and achieving its intended purpose. 

1. Please consider that the term telehealth is interchangeably with the term telemedicine (use reference doi: 10.1056/NEJMsr1503323.)

2. Line 63. Explain the acronym MAS at its first appearance

3. The MAs concept should be better explained. Eg. A MAS is a system composed of multiple interacting intelligent agents. The agents in a multi-agent system can be autonomous entities, such as software agents, robots, or humans, that operate in a shared environment and interact with one another to accomplish a common goal or set of goals.

4. Why MAS can be useful in telemedicine? Please consider that Telemedicine is per se a multi-agent system that involves multiple agents, such as patients, healthcare providers, and medical devices, working together to deliver healthcare services remotely. For instance, a patient can use a remote monitoring device to collect vital signs and transmit the data to a healthcare provider for analysis. (use doi: 10.1016/j.jbi.2020.103483.)

3. You should distinguish among different chronic pain problems. For example, telerehabilitation programs are adopted for fibromyalgia (10.3390/ijerph18042075). Furthermore, it was investigated that a patient-centred physical activity intervention, supported by health coaching and mobile health, to reduce care-seeking, pain and disability in patients with chronic low back pain (see and cite: DOI: 10.1186/s12891-019-2454-y ).

4. More importantly, telemedicine is also used for cancer pain management. See doi: 10.1093/pm/pnac128. for this aim, it is a valuable instrument (use doi: 10.3390/curroncol29080439)

5. telemedicine is also used for research aims (doi: 10.1002/cncr.30172.), also in cancer pain and through artificial intelligence methods (doi: 10.21873/invivo.13090.)

6. Having some tables that show the studies and the main findings could help with the reading of such a detailed text.

7. Expand paragraph 4. Telehealth Modalities. Use references doi: 10.1377/hlthaff.2018.05132.

8. Address MAS use and its application for potential telemedicine care models (hybrid processes; in person/remote consultations strategies: when, why ...). refer to: doi: 10.21873/invivo.13090.

9. Ethical issues should be better addressed (see and cite doi: 10.1007/s11606-017-4082-2)

10. About limitations, when you discuss the issue of technology, you must consider the required training and the role of caregivers (see and cite doi: 10.1080/0142159X.2020.1799959.; doi: 10.1200/EDBK_200141.)

Author Response

Date: March, 7, 2023

Manuscript number: clinpract-2235424

Title: Telemedicine, E-health, and Multi-Agent Systems for Chronic Pain Management

Dear Editor in Chief,

Thank you for your recent correspondence regarding our paper on Telemedicine, E-health, and Multi-Agent Systems for Chronic Pain Management, submitted to Clinics and Practice. Here is a corrected manuscript with a point-by-point response to comments from the Reviewers. The comments by the reviewers were of great value and each point had been addressed below.

Specifically, with regard to Reviewers' Comments to Author:

Reviewer 1 comments:

This paper highlights the benefits of telemedicine, telehealth, and e-health in pain management and healthcare services, especially in challenging circumstances. It also addresses the role of Multi-Agent Systems (MAS) in these fields, which is rapidly evolving as a solution to various real-world problems. MAS and information technology are used to enhance patient-centered palliative care and are considered critical technologies in managing chronic pain. However, the published research on their use in treating chronic pain is currently limited.

Although the paper provides a useful overview of the advantages and disadvantages of each technology, I have a few suggestions for how the text could be improved. In brief, some possibilities include making modifications to the language, adding additional information (e.g., references), clarifying certain points, or making the text more engaging and visually appealing. By implementing these suggestions, the text will become more effective in conveying its message and achieving its intended purpose.

  1. Please consider that the term telehealth is interchangeably with the term telemedicine (use reference doi: 10.1056/NEJMsr1503323.)

Thank you for the comment. The manuscript has been updated to note to readers that the two terms are very similar and are commonly used interchangeably.

  1. Line 63. Explain the acronym MAS at its first appearance

The acronym is now explained.

  1. The MAS concept should be better explained. Eg. A MAS is a system composed of multiple interacting intelligent agents. The agents in a multi-agent system can be autonomous entities, such as software agents, robots, or humans, that operate in a shared environment and interact with one another to accomplish a common goal or set of goals.

A slight addition was added towards the end of the first paragraph in section 2. Background about Multi-Agent Systems. This portion added serves to better explain what is providing the information to the autonomous agents. It is important to note that automatic or human input is what guides and directs these autonomous agents to work on their own and cross reference data from a patient’s entire history.

  1. Why MAS can be useful in telemedicine? Please consider that Telemedicine is per se a multi-agent system that involves multiple agents, such as patients, healthcare providers, and medical devices, working together to deliver healthcare services remotely. For instance, a patient can use a remote monitoring device to collect vital signs and transmit the data to a healthcare provider for analysis. (usedoi: 10.1016/j.jbi.2020.103483.)

This point is now stated in the Benefits and Limitations of Telehealth section discussing the missing component of telehealth which would be the autonomous agents of MAS that can easily put the data together.

  1. You should distinguish among different chronic pain problems. For example, telerehabilitation programs are adopted for fibromyalgia (10.3390/ijerph18042075). Furthermore, it was investigated thata patient-centred physical activity intervention, supported by health coaching and mobile health, to reduce care-seeking, pain and disability in patients with chronic low back pain (see and cite:DOI:10.1186/s12891-019-2454-y ).

These two papers have been utilized to show the dynamic approach of telehealth. These are appropriate to narrate the dynamic usage of telehealth in patient care.

  1. More importantly,telemedicine is also used for cancer pain management. Seedoi: 10.1093/pm/pnac128. for this aim, it is a valuable instrument (usedoi: 10.3390/curroncol29080439)

Telehealth’s utility in the management of cancer pain has been noted and talked about in the paper.

  1. telemedicine is also used for research aims (doi: 10.1002/cncr.30172.), also in cancer pain and through artificial intelligence methods (doi: 10.21873/invivo.13090.)

More information has been added based on the first article. The second article listed, however, does not seem to provide a clearcut usage of telehealth and has unclear results.

  1. Having some tables that show the studies and the main findings could help with the reading of such a detailed text.

 Table 1: Benefits and limitations of Telehealth was added.

  1. Expand paragraph4. Telehealth Modalities. Use references doi: 10.1377/hlthaff.2018.05132.

           This portion was expanded and discussed the studies describing improved diagnosis.

  1. Address MAS use and its application for potential telemedicine care models (hybrid processes; in person/remote consultations strategies: when, why ...). refer to:doi: 10.21873/invivo.13090.

MAS is now described in a hybrid scenario and importantly discusses how it can be paired with E-health systems to provide the best patient care. The addition of MAS to these systems allows for decision making processes on when to use it. The segment also discusses why as it pertains to usage to produce more efficient results.

  1. Ethical issues should be better addressed (see and citedoi: 10.1007/s11606-017-4082-2)

Ethical issues for a physician are now discussed, referring to confidentiality, fidelity, continuity of care, and so on. The role of the physician is described as a user of telemedicine to foster a better patient environment.

  1. About limitations, when you discuss the issue of technology, you must consider the required training and the role of caregivers (see and citedoi: 10.1080/0142159X.2020.1799959.;doi: 10.1200/EDBK_200141.)

 These limitations are discussed and added into the section about telehealth. The two articles provide firm backgrounds to suggest the problems with learning the different telehealth tools.

Reviewer 2 comments:

Thank you very much for giving me the opportunity to give my opinion and make my remarks on this review. It’s a very important topic for the current time, but even more important for the future of the medical management. There is not enough information in the literature which describes the plans and actions of using all these tools of information technology and communication technology in the future. There is no doubt that the precise and correct use of all these tools have reals benefits already cited by the authors.

Below, I am describing some remarks with the aim of improving the quality of this review.

  1. The main objective of this paper is to discusses why telemedicine, E-health, and MAS are the most critical novel technologies for providing health care and managing chronic pain. I should emphasize that the main purpose of this review should be the description how to use these new technologies in the management of chronic pain. Unfortunately, the authors do not adequately respond to this specific objective of the review, which should focus on the application of these technologies on the screening, obtaining accurate information, adequate treatment and follow-up of patients with chronic pain.

While the direct methods of telehealth devices are not explicitly discussed in the literature, many usages and results have been talked about. The paper has multiple new additions of different telehealth devices or programs and their corresponding results and benefits for patients. They primarily span the usage in chronic pain management but also discuss the widespread usability of telehealth in healthcare.

  1. The authors describe in a very general way the application of these technologies in the field of medicine. My suggestion is that this description should be done in a very concrete way and with examples cited in different publications. It would be much more understandable to describe the use of these modern technologies in the management of patients with chronic pain in the way to answers the question of the primary objective of this review.

Multiple studies were added into the section on telehealth to better show the improvements and results of telehealth for patients. Multiple studies on chronic pain management with telehealth tools. The topics discuss cancer pain management and multiple musculoskeletal pain problems. Many of these studies discuss the obvious benefits and positive effects of these programs.

  1. The authors describe in very few words the negative sides of these technologies. It would be much more interesting to further detail the problems related to the use of these modern technologies, such as the lack of social and human contact, the leak of secret medical information, the other informatic technology problems, the greater difficulty of using these tools for disabled and depressed patients.

More ethical dilemmas and concerns for health care workers were added to the manuscript, discussing the fidelity, confidentiality, and attention to care for health care workers to be aware of. Furthermore, it discusses the potential for negative side effects of decreased interaction as well as the implications for mental health problems such as anxiety and loneliness. In the case of depressed patients, the telehealth was discussed as a potential downside in terms of human interaction and ability to overcome feelings of anxiety and loneliness. However, for disabled patients, other than direct hindrances of using technology, telehealth has commonly be cited as a form of direct benefit and aide to these patients.

Sincerely,

Sahar Shekoohi, PhD

Post-Doctoral Fellow

Louisiana State University Health Sciences Center at Shreveport, Department of Anesthesiology, Shreveport, LA, 71103, USA

Sahar.Shekoohi@Lsuhs.edu

Alan David Kaye, MD PhD
Professor, Louisiana State University Health Science Center at Shreveport, Departments of Anesthesiology and Pharmacology, Toxicology and Neurosciences, Pain Fellowship Program Director, alan.kaye@lsuhs.edu

Reviewer 2 Report

Dear Editor,

Thank you very much for giving me the opportunity to give my opinion and make my remarks on this review. It’s a very important topic for the current time, but even more important for the future of the medical management. There is not enough information in the literature which describes the plans and actions of using all these tools of information technology and communication technology in the future. There is no doubt that the precise and correct use of all these tools have reals benefits already cited by the authors.

Below, I am describing some remarks with the aim of improving the quality of this review.

1.       The main objective of this paper is to discusses why telemedicine, E-health, and MAS are the most critical novel technologies for providing health care and managing chronic pain. I should emphasize that the main purpose of this review should be the description how to use these new technologies in the management of chronic pain. Unfortunately, the authors do not adequately respond to this specific objective of the review, which should focus on the application of these technologies on the screening, obtaining accurate information, adequate treatment and follow-up of patients with chronic pain.

2.       The authors describe in a very general way the application of these technologies in the field of medicine. My suggestion is that this description should be done in a very concrete way and with examples cited in different publications. It would be much more understandable to describe the use of these modern technologies in the management of patients with chronic pain in the way to answers the question of the primary objective of this review.

3.       The authors describe in very few words the negative sides of these technologies. It would be much more interesting to further detail the problems related to the use of these modern technologies, such as the lack of social and human contact, the leak of secret medical information , the other informatic technology problems, the greater difficulty of using these tools for disabled and depressed patients.

Author Response

(The authors gave the same response as above.)

Round 2

Reviewer 2 Report

Dear Editor,

I read the changes made by the authors regarding my remarks made in the initial version. The authors have made good progress in improving the quality of the manuscript. In my opinion, the problem consist again in the first point of my initial remarks.

Since the authors aim to reflect the use of telemedicine, MAS and E health on chronic pain management, this should be the core part of this manuscript. The chronic pain management problem should be presented in a larger and more specific way from the beginning of this article.

However, the topic of this manuscript remains of a very special importance for the future of medicine.

Author Response

Date: March, 12, 2023

Manuscript number: clinpract-2235424

Title: Telemedicine, E-health, and Multi-Agent Systems for Chronic Pain Management

Dear Editor in Chief,

Thank you for your recent correspondence regarding our paper on Telemedicine, E-health, and Multi-Agent Systems for Chronic Pain Management, submitted to Clinics and Practice. Here is a corrected manuscript with a point-by-point response to comments from the Reviewers. The comments by the reviewers were of great value and each point had been addressed below.

Specifically, with regard to Reviewers' Comments to Author:

Reviewer comments:

I read the changes made by the authors regarding my remarks made in the initial version. The authors have made good progress in improving the quality of the manuscript. In my opinion, the problem consist again in the first point of my initial remarks.

Since the authors aim to reflect the use of telemedicine, MAS and E health on chronic pain management, this should be the core part of this manuscript. The chronic pain management problem should be presented in a larger and more specific way from the beginning of this article.

However, the topic of this manuscript remains of a very special importance for the future of medicine.

Thank you for the comments. Here is a summary to discuss our edits in each section.

  1. Introduction - We added more emphasis on the potential of these systems for chronic pain as well as their existence but limited use as seen in the literature.
  2. Background about multi-agent systems - We did not see a good place to add direct chronic pain info as this section is solely based on explaining MAS.
  3. Multi-Agent Systems and Telehealth - We added an entire paragraph focusing on the implementation and advantages of these two systems in chronic pain management to start discussing it earlier in the manuscript. 
  4. Telehealth modalities - Once again, we added more information on direct ability to ease patient's lives through telehealth chronic pain management and used studies showing its positive effects.
  5. Benefits and Limitations of Telehealth - We added small sentences to further discuss parts but found most of the stuff that we added in the previous comments covered this section.
  6. Multi-Agent Systems and E-health - We added another article discussing E-health in chronic pain management. We felt that this section already had sufficient emphasis on chronic pain management.

Sincerely,

Sahar Shekoohi, PhD

Post-Doctoral Fellow

Louisiana State University Health Sciences Center at Shreveport, Department of Anesthesiology, Shreveport, LA, 71103, USA

Sahar.Shekoohi@Lsuhs.edu